# Fused Deposition Modeling of Poly (Lactic Acid)/Walnut Shell Biocomposite Filaments—Surface Treatment and Properties

**Xiaohui Song [1,2,3], Wei He [1,*], Shoufeng Yang [3,4] 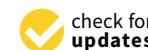, Guoren Huang [1] and Tonghan Yang [1]**

[1]   College of Chemistry and Chemical Engineering and Guangxi Key Laboratory of Processing for Non-ferrous Metallic and Featured Materials, Guangxi University, Nanning 530004, China; songxiaohui2010@163.com (X.S.); 15951701683@163.com (G.H.); yangthan199@163.com (T.Y.)

[2]   College of Mechanical Engineering, Guilin University of Aerospace Technology, Guilin 541004, China

[3]   Faculty of Engineering and Environment, University of Southampton, Southampton SO17 1BJ, UK; s.yang@soton.ac.uk

[4]   Member of Flanders Make, Department of Mechanical Engineering, KU Leuven, P.O. Box 2420, 3001 Heverlee, Leuven, Belgium

*   Correspondence: wei_he@gxu.edu.cn

**Abstract:** This paper presents the study of the properties of objects that were fabricated with fused deposition modeling technology while using Poly (lactic acid)/Walnut shell powder (PLA/WSP) biocomposite filaments. The WSP was treated while using NaOH followed by silane. The infrared spectrum of treated and untreated WSP was characterized. The result was that thermal and mechanical properties could be improved by adjusting the concentration of silane. The experimental results showed: the surface compatibility between WSP and PLA was dramatically improved through treatment with KH550. The crystalline, thermal gravity, and thermal degradation temperatures of biocomposite with untreated WSP were improved from 1.46%, 60.3 °C, and 239.87 °C to 2.84%, 61.3 °C, and 276.37 °C for the biocomposites with treated WSP, respectively. The tensile, flexural, and compressive strengths of biocomposites were raised each by 8.07%, 14.66%, and 23.32%. With the determined silane concentration, PLA/10–15 wt.% treated WSP biocomposites were processed and tested. The results showed that the tensile strength was improved to 56.2 MPa, which is very near to that of pure PLA. Finally, the porous scaffolds with controllable porosity and pore size were manufactured.

**Keywords:** fused deposition modeling; biocomposite; poly (lactic acid)-matrix; walnut shell; surface treatment

## 1. Introduction

Researchers have focused more on biodegradable and sustainable materials research due to the ever-growing global energy crisis and ecological risks. Poly (lactic acid) (PLA), derived from renewable resources (e.g., corn, wheat, or potato) [1], is a biodegradable, recyclable, and compostable thermoplastic with good biocompatibility and processibility. Whereas the slow degradation rate, high cost, hydrophobicity, and lack of reactive side-chain groups have limited its use in the biomedical field [2]. Natural fibers are being used currently as reinforcement for PLA, such as seed (e.g., cotton and milk weed), fruit shell (e.g., walnut, almond and peanut), bast (e.g., flax, lamp and jute), leaf (e.g., sisal and banana), and grass/cane/reed fibers (e.g., bamboo) [3]. Natural fibers possess a lot of advantages, including a high degree of flexibility, low cost, light weight, non-toxic, and high specific modulus. Additionally, they are biodegradable and recyclable in contrast to traditional synthetic fibers [4].

As a kind of natural fiber, Walnut Shell (WS) is a very common by-product of walnut food. A health claim that was authorized by Food and Drug Administration (FDA) indicated that walnuts could reduce the risk of heart disease [5]. It takes up 67% of the total weight of a walnut. According to the data from the United States Department of Agriculture, China's walnut production was about 850,000 metric tons in the marketing year of 2018/2019 [6]. However, although a small amount of WS was used as adsorbent [7,8], most of them were scraped as waste. Like other natural fibers, WS consists of cellulose, hemicelluloses, lignin, and a small part of wax, oil, and other impurities on its surface [9]. These impurities prevent WSP from reacting with the hydrophobic polymer matrix, which results in poor interfacial compatibility, which further leads to the debonding and formation of voids [10,11]. Normally, this issue could be partially solved through physical and chemical modifications [12]. Many chemical treatment methods have been reported in prior literatures, such as alkali [4], silane coupling [13], maleation [14], and so on. Beyond that, some agents have also been used to improve the interfacial compatibility, such as montmorillonite [15], metakaolin [16], and talc [17].

Several researchers have conducted valuable work on the polymer/WSP composites. Ayrilm is et al. [18] introduced 40–60% WS into Polypropylene (PP) and found that the bending and tensile modulus of composites were significantly improved with the increase of filler content. Guru et al. [19] fabricated a WSP/urea–formaldehyde particle board with a tensile strength of 3.8 MPa. Sancak et al. [20] found that a 40% addition of WSP greatly enhanced the tensile and flexural strength of WS/Polyester composites. In addition to these papers, the mechanical properties of Epoxy/WS composites were reported in some papers [21,22]. Sasari et al. [23] evaluated the effect of WS on the tensile property of the Starch/WSP composite. Liu [24] investigated the effect of WS and KH550 on the properties of PLA/WS composite. However, these WS/polymer composites were all prepared by traditional methods, including injection molding and compression molding [25,26], which would limit its application in some parts with a complex internal and external structure.

Fused Deposition Modeling (FDM) has advantages over traditional technologies of reducing product development times and free fabricating complex-shaped parts [27]. FDM is currently one of the most popular Additive Manufacture (AM) techniques, due to its low cost and easy operation, along with its wide application in bio (polymer) and recent biocomposites [28], including PLA/wood composites [25,29], PLA/kraft pine lignin [26], PLA/jute and flax strands [30], PLA/basalt composites [31], etc. Even though, FDM of polymer/natural-fibers composites was rarely reported in the literature.

The main goal of this study is to fabricate relatively cheap PLA biocomposite filaments with WSP incorporated such that the objects that were obtained with FDM technology have acceptable mechanical properties. Another goal is to investigate the effect of alkali and silane treatment on the rheological behavior and the thermal properties of biocomposites. The WSP was modified by NaOH/2-8 wt.% KH550, and it was tested with Fourier-Transform Infrared Spectrometer (FTIR) to investigate its modified chemical groups. Subsequently, the biocomposite powders were extruded into filaments and further fabricated with the Fused deposition modeling (FDM) technology. The thermal and mechanical properties were investigated. The microstructure of flexural fractured surface was characterized. With the determined silane concentration, PLA/WSP (10 wt.% and 15 wt.%) composites were prepared and their properties were tested. Additionally, the porous scaffolds were successfully manufactured.

## 2. Experiments

### 2.1. Materials and Methods

#### 2.1.1. PLA

Poly (lactic acid) (PLA) 2002D was a medical-grade polymer powder that was purchased from Nature Works LLC, USA. According to the data sheet, it has a specific gravity of 1.24 g/cm$^{-3}$ and a molecular weight of 100,000 g/mol [32]. PLA has a glass transition temperature of 60.9 °C, a melting point temperature of 152.1 °C, and a melt flow index of 10 g/10 min.

2.1.2. Walnut Shells and Surface Treatment

The Walnut Shell (WS) used in this research was obtained from a local planter in the south of Shaanxi, China. Prior to the use, they were manually chosen to remove the qualitatively inferior items, cleaned with distilled water to remove the dust and impurities, and then dried in an air dry oven at 85 ± 2 °C for 24 h to reduce the moisture. The dried materials were grounded into powder while using a high-speed rotary cutting mill (CS-700Y, COSUAI).

The surface treatment of the Walnut Shell Powder (WSP) was carried out with alkali (NaOH), followed by silane, to reduce moisture sensitivity and enhance the interface compatibility with the matrix [33]; Figure 1 shows the treatment procedure. Firstly, the WSP was soaked in a solution of aqueous alkaline (NaOH) for 3 h at room temperature. The ratio between WSP and solution was 1:20 (wt/wt). After the NaOH treatment, in order to remove any traces of NaOH from the surface of WSP, deionized water was used to wash the WSP several times, and acetic acid solution was added to neutralize the solution. The NaOH treated WSP was dried at 85 ± 2 °C for 48 h. Secondly, silane coupling agent (3-amino propyl triethoxy silane, KH550) was used to further treat WSP. Prior to the silane treatment, the acetic acid was added in deionized water to make the pH 3.5–4. Afterwards, 2 wt.%, 4 wt.%, 6 wt.%, and 8 wt.% KH550 was dissolved in deionized water, respectively, for hydrolysis. The solution was continuously stirred while using a magnetic stirring apparatus for 30 min to hydrolyze the silane completely. The NaOH treated WSP was soaked in the prepared silane solution and continuously stirred for 8 h at room temperature. After the treatment, the WSP was washed with deionized water for several times to remove the silane residues from the WSP surface and dried in a laboratory oven at 85 °C to achieve a moisture content of 0–1% measured using a scale with an error of 0.001.

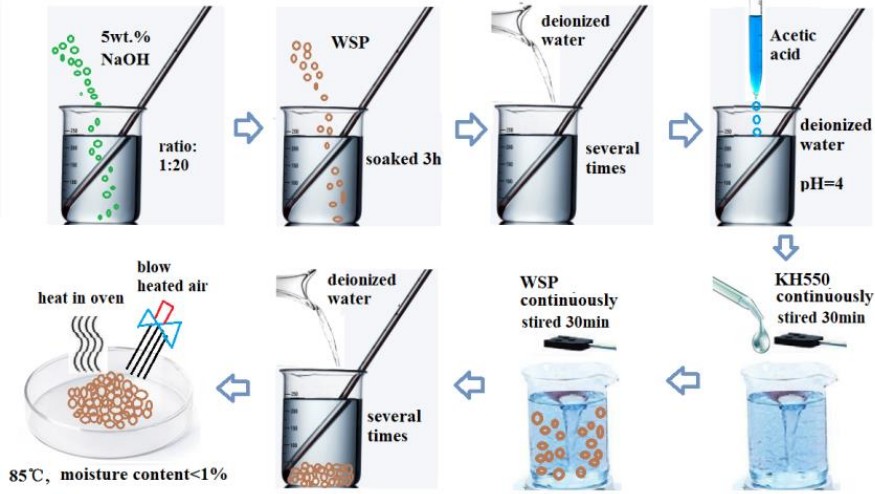

**Figure 1.** The surface treatment process of Walnut Shell Powder (WSP).

Figure 2a,b show the micrograph of WSP treated and untreated with NaOH/silane, respectively. The micrographs were examined while using a scanning electron microscopy (SEM, HITACHI SU8020 system). The WSP particles in both two figures display an irregular shape. There were micro pores that were dispersed on the surface of WSP with the shape of an oval and an average size of 1–2 μm (white rectangle). In Figure 2a, smaller WSP particles had the tendency to agglomerate together. The incorporation of the NaOH/silane into the WSP, the surface of particles was rougher than untreated one, and the phenomenon of agglomeration was removed. The results showed that the treated WSP is more compatible with the PLA matrix.

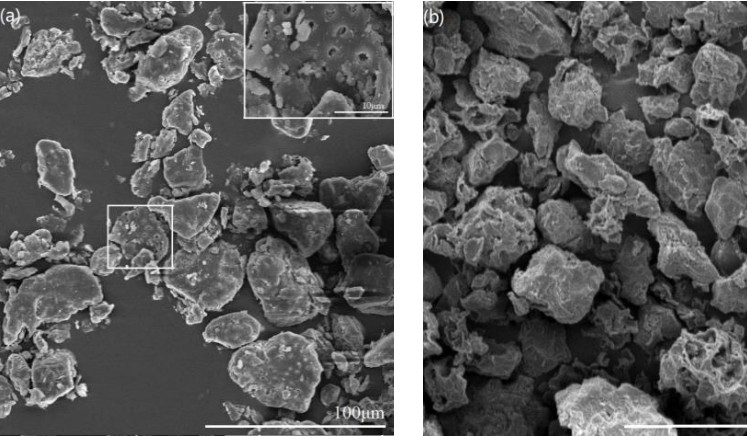

**Figure 2.** The morphology of (**a**) untreated WSP and (**b**) NaOH/silane treated WSP.

## 2.2. Composite Preparation

### 2.2.1. Powder Biocomposites Preparation

The particle size ratio between the matrix material and the filler should be about 10:1 or over, which is beneficial to the homogeneity of the mixture [34]. Before preparation, the PLA powder was sieved with a 35-mesh screen (~500 μm), and the WSP was sieved while using a 300-mesh screen (~50 μm). Figure 3 shows the particle size distributions of PLA and WSP powder. Most of the PLA powder has particle size between 300–700 μm and the particle size distribution of WSP is 20–45 μm. Their distributions satisfied the required size ratio. Subsequently, five types of biocomposite powders, PLA/5 wt.% untreated WSP and PLA with 5 wt.% WSP that was treated with NaOH/2, 4, 6, 8 wt.% KH550 (abbreviated as PLA-untreated WSP, PLA-Na treated WSP, PLA-*%NaSi treated WSP) were dry-milled for 24 h using a mechanical planet ball grinder machine (QM-3SP4, Nanjing Yifan Apparatus Co. Ltd., Nanjing, China). Two kinds of Zirconium balls with an average diameter of 5 mm and 20 mm, respectively, were added to improve the grinding efficiency and homogeneity of the mixture [35]. The weight ratio of ball and material was 1:1.

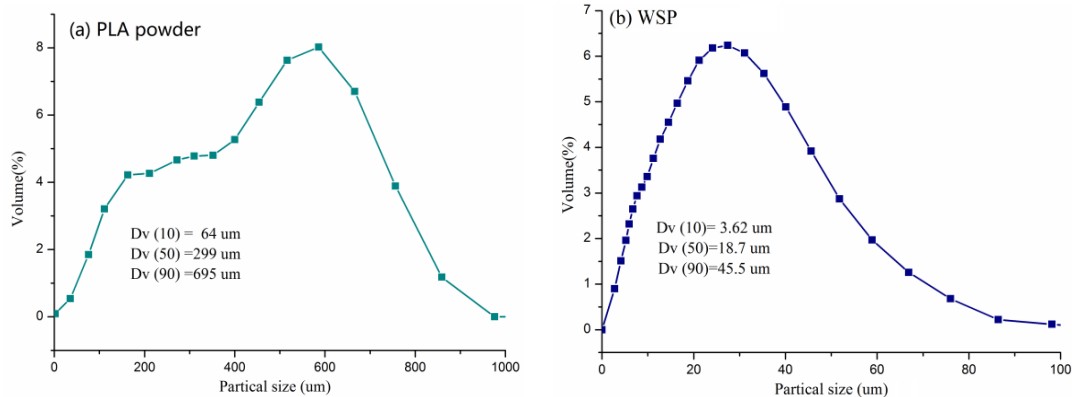

**Figure 3.** The particle size distribution of: (**a**) poly (lactic acid) (PLA) powder, (**b**) WSP.

### 2.2.2. Preparation Filaments for FDM

A single screw extruder was chosen to extrude the filaments from the biocomposite powder in order to prepare the new biocomposite filaments served on FDM. The screw diameter of the extruder was 20 mm and the ratio of length and diameter was 25. The filaments served in FDM need to satisfy a specific diameter (~1.75 mm). The outlet swell effect during extrusion process need to be considered [34]. Taking this into account, the variable parameters, including barrel temperature, screw speed and die

nozzle diameter, were examined. According to the thermal properties of materials and after the results of trial and error, the parameters were determined as: 20 rpm for the screw speed, 160 °C for the barrel temperature, and 1.5 mm for the die diameter. Eight types of filaments were fabricated, including pure PLA, PLA/untreated WSP, PLA/2% NaSi treated WSP, PLA/4% NaSi treated WSP, PLA/6% NaSi treated WSP, PLA/8% NaSi treated WSP, PLA/10 wt.% treated WSP, and PLA/15 wt.% treated WSP. Figure 4a,b show the extruded pure PLA filament and PLA/6% NaSi treated WSP filament, respectively, with a diameter within the range of 1.72 ± 0.3 mm, which were inspected while using a digital vernier caliper with a range of 0–25 mm and an error range of 0.001 mm. This type of filament can be used for the FDM.

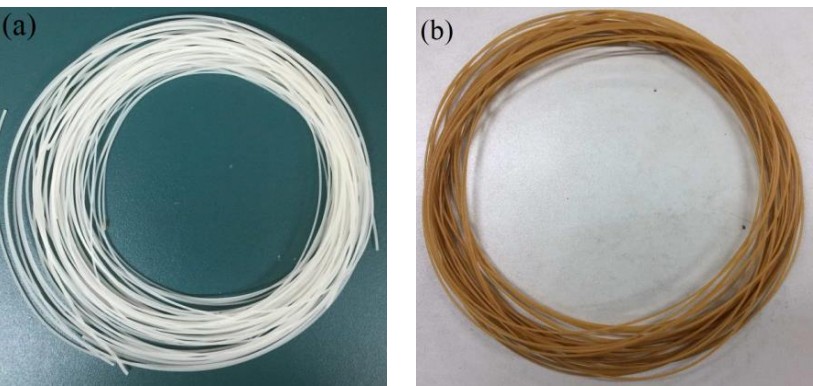

**Figure 4.** (**a**) Pure PLA filament, (**b**) PLA/WSP filament.

### 2.2.3. Fabrication of Specimens on FDM

After inspection, the filaments were spooled and then directly loaded into the commercial FDM machine, respectively. The Allct Yinke FDM machine (with a slicer of Allct software) that was developed by the Allct technology company, Wuhan, China, was used. Prior to the manufacturing of specimens, an orthogonal experiment was carried out to determine the processing parameters. The processing parameters were as follows: 50 mm/s of printing speed V, 210 °C of nozzle heater temperature T, 0.06 mm of layer thickness H, 100% fill ratio, 0.4 mm of shell thickness, and 40 °C for the platform temperature and linear filling mode. For each kind of biocomposite material, flexural, tensile, and compressive specimens were manufactured, respectively. Based on the GBT 9341-2008 standard, four rectangular-shaped specimens with a dimension of 80 mm × 10 mm × 4 mm were prepared for a flexural test. According to the ASTM-638 standard, four dumbbell-shaped samples with 63.5 mm of length, 9.53 mm of width, and 3.2 mm of thickness were fabricated for a tensile test. Five cylindrically shaped specimens with a dimension of 10 mm in diameter and 12 mm in height were manufactured for compression testing. Figure 5 shows the as-FDM PLA and PLA/4%NaSi specimens that were used for the mechanical test.

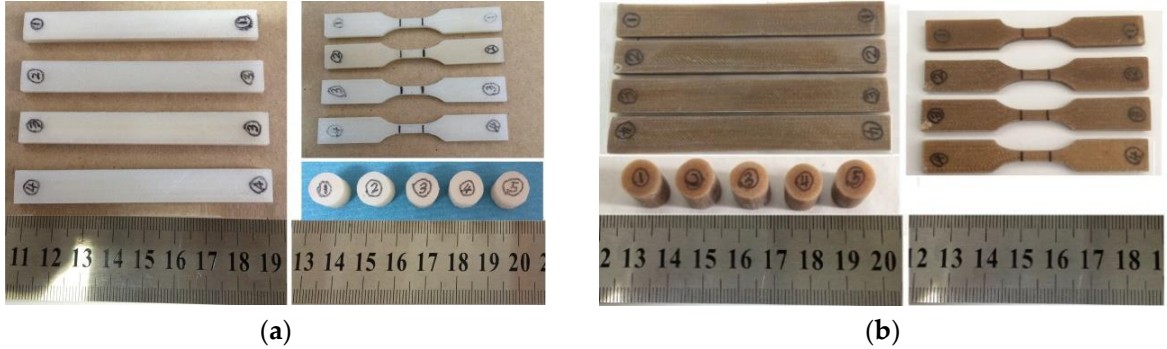

(**a**)          (**b**)

**Figure 5.** The mechanical test specimens of (**a**) pure PLA and (**b**) PLA/4% NaSi treated biocomposites fabricated on a Fused Deposition Modeling (FDM) setup.

### 2.3. Characterization Techniques

#### 2.3.1. Thermal Characterization of Biocomposites Powder

The thermal properties were investigated while using the Differential Scanning Calorimeter (DSC), Labsys Evolution 1600 (Setaram, France). Biocomposites of approximately 25 mg were placed into ceramic pans and the tests were performed in a dry nitrogen atmosphere (flow rate of 20 mL/min). The samples were submitted to a heating program (heating from 30 °C to 200 °C at the rate of 10 °C/min, holding for 3 min at 200 °C, and then cooling from 200 °C to 30 °C at the rate of 10 °C/min). The glass transition temperature ($T_g$) was obtained from the inflection point of the specific heat capacity; the cold peak temperature ($T_{cc}$) and the melting temperature ($T_m$) were taken at the peak of the crystallize endotherms and the melting endotherms, respectively. The enthalpies of fusion ($\Delta H_m$) and the enthalpies of crystallization ($\Delta H_{cc}$) were obtained from the areas under the peaks. An Indium sample was used as the calibration standard. Crystallinity ($X_C$ (%)) was calculated with the following equation [36]:

$$X_C = \left( \frac{\Delta H_m - \Delta H_{cc}}{w \Delta H_m^0} \right) \times 100\% \tag{1}$$

where, $\Delta H_m$ is the experimental melting, $\Delta H_{cc}$ is the experimental cold crystallization enthalpy, $\Delta H_m^0$ is the enthalpy of melting of 100 percent crystalline PLA (considered as 93 J/g), and w is the PLA weight fraction in the biocomposite [37,38].

The thermal stability of the PLA/WSP biocomposites was determined while using a Thermo Gravimetric (TG) analyzer, Labsys Evolution (Setaram, France), with a thermo balance sensitivity of 0.2 μg. The temperature calibration of about 25 mg samples was done at 30–500 °C, at the rate of 10 °C/min, with a nitrogen purge flow of 25 mL/min, while using ceramic crucibles.

#### 2.3.2. Fourier-Transform Infrared Spectroscopy (FTIR)

The surface functional groups and structure were investigated using FTIR (Thermo Nicolet AVATAR FTIR 360 instrument). The FTIR spectra of the raw materials and biocomposites were recorded from a wave number 400–4000 cm$^{-1}$, at a spectral resolution of 4 cm$^{-1}$. Before the test, a 2–3 mg sample and 200–300 mg KBr were ground together while using an agate crucible and dried sufficiently; and then made into a tablet on a tablet press. The spectrum of dry KBr served as the background and it was subtracted from the experimental spectrum.

#### 2.3.3. Rheological Behavior

The successful FDM of a new particle filled polymer filament strongly depends on the rheological behavior of the composites materials. According to the standard ASTM D1238-73, the Melt Flow Index (MFI) of PLA/WSP biocomposites was measured on a melt flow tester (Tiansu cablication, Shenzhen, China) with a capillary die with a standard dimension (diameter of 2.0955 ± 0.001 mm). The test was carried out by maintaining a cylinder temperature at 160 °C and a load of 3.8 kg. The results were recorded in grams per 10 min (five times per sample).

#### 2.3.4. Mechanical Properties

The tensile properties of standard specimens were tested while using a Universal Testing Machine (Germany Zwick Roell, 2 KN) with a test rate of 0.05 mm/s and preload of 0.1 N. The compression testing and flexural testing were performed with a Universal testing machine (Model8800, Instron, Canton, MA, USA). The crosshead speed for compression testing was 1 mm/min and 0.5 mm/min for a three-point bending test. The preload for both was 0.1 N. For each sample, four measurements were made to obtain the average value of strength and strain.

### 2.3.5. Morphological Characterization

The morphological analysis allows for observing the surface of the fracture surface of prepared biocomposites. Morphological studies were performed while using Scanning Electron Microscopy (SEM, HITACHI SU8020 system) that was equipped with an energy dispersive spectrometer (EDS) analysis system. Prior to the scanning, all of the materials were gold coated.

### 2.3.6. Porosity

The porosity of the scaffold can be calculated by this equation [39].

$$Porosity = 1 - \frac{\rho_0}{\lambda_{PLA} \times \rho_{PLA} + \lambda_{WSP} \times \rho_{WSP}} \tag{2}$$

where $\lambda_{PLA}$ and $\lambda_{WSP}$ are the weight ratio of PLA and WSP in biocomposites; $\rho_{PLA}$ and $\rho_{WSP}$ are the theoretical densities (g/cm$^3$) of PLA and WSP, $\rho_{PLA}$ is equal to 1.24 g/cm$^3$ provided by the manufacturers, $\rho_{WSP}$ equals to 1.079 g/cm$^3$ measured with pycnometer method; and, $\rho_0$ is the apparent density obtained through.

$$\rho_0 = m_0 / V_0 \tag{3}$$

where $m_0$ is the sample mass (g) measured with a scale with accuracy of 0.001 g and $V_0$ is the sample volume (mm$^3$) obtained by a Vernier caliper. The average density of four scaffolds was taken as the density of the PLA/WSP biocomposite.

## 3. Results and Discussions

### 3.1. FTIR Analysis of Untreated and Treated WSP

Figure 6 shows the FTIR spectrums of untreated and treated WSP. The spectrum of untreated WSP conforms to that of the natural fibers proved in references [7,40]. The band at 1744 cm$^{-1}$, contributing to the stretching vibration in carbonyl C = O groups, vanished after the introduction of NaOH into WSP, indicating that the hemicelluloses, along with ash, oil, and other impurities had been removed. The band of lignin at 1038 cm$^{-1}$ cleaved into several small peaks after the treatment with NaOH. This implied that the number of lignin increased due to the disappearance of hemicellulose.

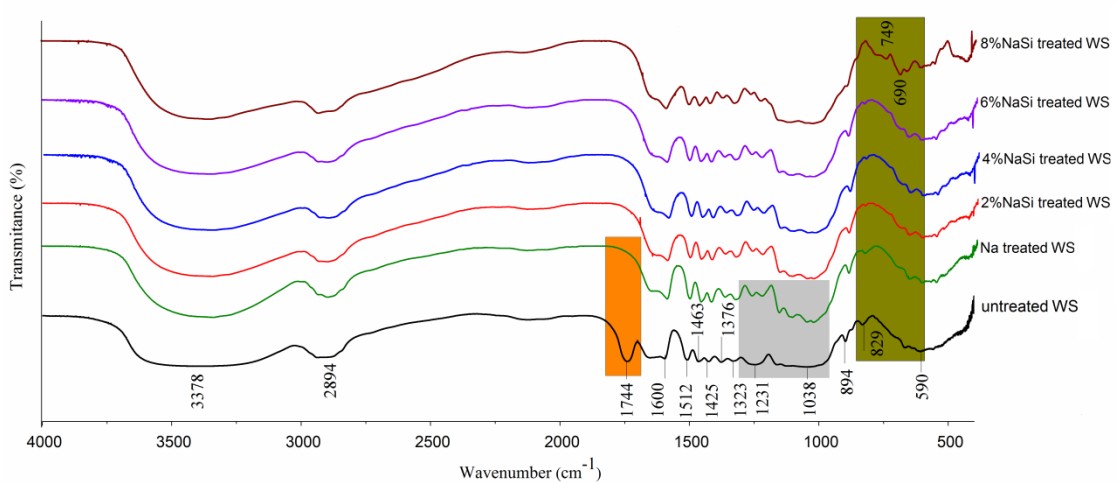

**Figure 6.** The Fourier-Transform Infrared Spectra (FTIR) spectra of WSP treated with NaOH/KH550. The orange box highlights the removing of hemicelluloses. The gray box highlights the cleavage of a lignin peak and the dark green box highlights the condensation reaction and auto polymerization.

After KH550 modification, the peak at 829 cm$^{-1}$ due to the C-H out-of-plane bending vibration faded away when the silane concentration was increased. Meanwhile, the-Si-C- symmetric stretching

band presented at 749 cm$^{-1}$, indicated that a condensation reaction had happened between the WSP (-C-H-) and silane coupling agent (-Si-O-H-). A downward peak that was relevant to the -Si-O-Si symmetric stretching showed up at 690 cm$^{-1}$, because of the auto polymerization among the silanol groups. The auto polymerize deposited on the surface of WSP, went into the micro pores of WSP and then formed an intertangling effect. The condensation reaction and the auto polymerization can improve the surface compatibility between the polymer matrix and the WSP.

### 3.2. Rheological Behavior

Figure 7 shows the rheological behavior of PLA/WSP biocomposites. It can be observed from Figure 7a that the melt flow index (MFI) is increasing with the increase of screen size, indicating that smaller WSP particles are beneficial for homogenously mixing with PLA. When the mesh size equals to 300 mesh (~50 μm), the MFI climbs up to a value of 10.58 g/10 min. The surface area of smaller particles is higher, which improves the heat transfer and decreases the heat stability [41].

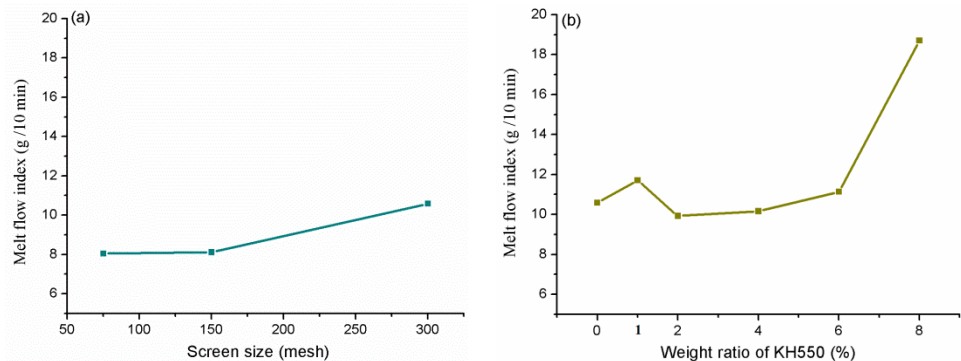

**Figure 7.** The relationship between Melt Flow Index (MFI) values and: (**a**), screen size and (**b**), weight ratio of KH550.

Figure 7b gives the relationship between MFI and the concentration of KH550, where "1" on the X axis represents the MFI of biocomposite with the NaOH treated WSP. Alkali treatment raised MFI of PLA/WSP biocomposites from 10.58 g/10 min to 11.7 g/10 min through the removal of the hemicelluloses and other impurities. The MFI of PLA/WSP biocomposites was dramatically increased after a small drop after silane was introduced on the NaOH treated WSP, and reached its highest value of 18.71 g/10 min. The higher MFI was probably due to a significant amount of auto polymerization silane molecules, which contain lighter weight molecules, resulting in the easy flow of biocomposites [42].

### 3.3. Thermal Properties Analysis of Untreated and Treated PLA/WSP Biocomposites

Figure 8a shows the DSC curves of PLA/WSP biocomposites and Figure 8b gives their crystalline values. It can be seen that all of the PLA/WSP composites experienced two endothermic and one exothermic phase transformation. The first endothermic inflection (black arrows) corresponds to the glass transition temperature (Tg, °C), while the second one represents the melting temperature (Tm, °C). The exothermic peak refers to the cold crystallization temperature (Tcc, °C) of PLA chains. Although changes of these transition temperatures are small, the effects of the WSP and the treatment on the thermal property of PLA/WSP biocomposites are obvious. The area of fusion and the area of crystallization become almost equal to each other, which causes the crystallinity (Xc) of the PLA/WSP biocomposites to become smaller (see the Equation (1)).

With the addition of WSP, the Tcc of the PLA/WSP biocomposite (sample 2, 115.2 °C) is higher than that of the pure PLA (sample 1, 113.1 °C). Whereas Tg, Xc, and Tm show opposite results with lower values of 59.6 °C, 1.46%, and 149 °C, when compared to pure PLA (61.9 °C, 1.8%, and 155.4 °C). This result indicated that the walnut shell did not effectively play the role of nucleating agent and it decelerated the crystallisation process of PLA, thus slowing down its crystal growth rate and decreasing

the Tcc. The WSP has a high amount of ash, oil, wax, and other impurities [43]. Under this situation, the molecular chain of WSP was loose-packing and had more mobility. The loose-packing WSP resulted in loose-packing composites when compounded into the PLA matrix. These impurities led the WSP particles to partially aggregate and hamper the interaction between cellulosic hydroxyl groups of the WSP and the PLA matrix, leading to the poor interfacial compatibility. Meanwhile, WSP particles with lightweight molecules might lubricate the PLA matrix [44], which enhances the mobility of PLA molecular chains and increasing the free volume of molecular chains, which contributed to the lower Tg, Xc and Tm.

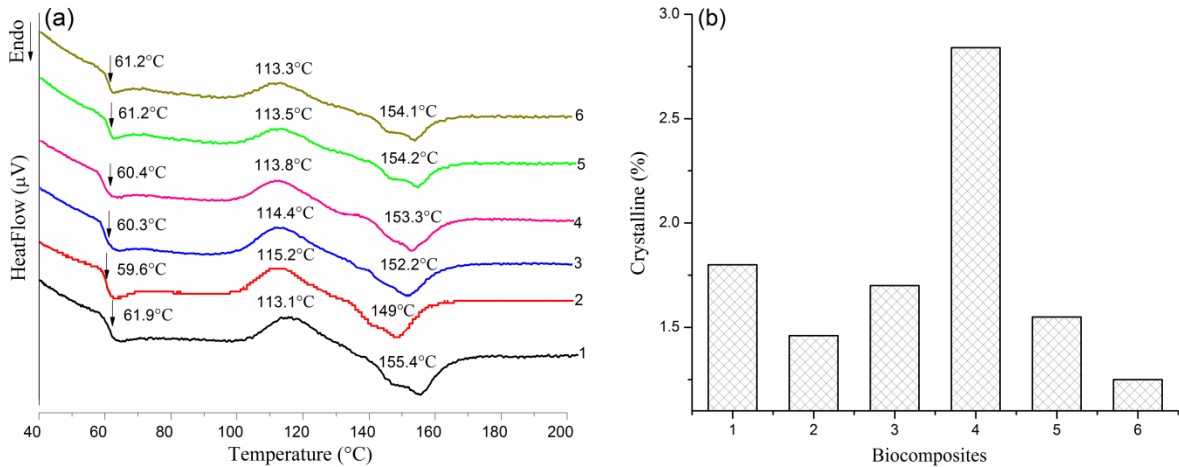

**Figure 8.** (**a**) The Differential Scanning Calorimeter (DSC) curves; 8 (**b**) the crystalline of PLA/WSP biocomposites: 1 PLA; 2 PLA/untreated WSP;3 PLA/2%NaSi treated WSP; 4 PLA/4%NaSi treated WSP; 5 PLA/6%NaSi treated WSP; and, 6 PLA/8%NaSi treated WSP.

The incorporation of the alkali and the silane increased the Tg and Tm, but decreased the Tcc of PLA/WSP biocomposites. With the increase of the KH550 concentration, the Tcc of PLA/WSP biocomposite showed a downward trend and slowly dropped to 113.3 °C (sample 6); while its Tg and Tm climbed steadily and reached the high points of 61.2 °C and 154.2 °C (sample 5), they then kept the level, regardless of the increase of the silane content. Firstly, alkali effectively removed the hemicelluloses and impurities from the WSP surface (Figure 6), which resulted in a better packing of cellulosic chains in the WSP. Secondly, the chemical bond -Si-C- between the WSP and silane acted as a bridge between the PLA matrix and the WSP fillers [24], resulting in a better surface compatibility. The deposition of auto polymerized siloxane on the micro pores and surface of WSP further functioned as the a mechanical interlock and enhanced the surface compatibility [45]. The chemical reaction and physical deposition both densified the molecules of PLA and limited the molecular mobility, speeding up the crystal growth rate, and accelerating the crystallisation process of PLA. This implied a lower crystal temperature and higher Tg and Tm, indicating that the NaOH/KH550 apparently improved the interfacial compatibility between PLA and WSP.

The Xc of PLA/WSP composites performed in a little complex but more dramatic way: it rocketed sharply to 2.84% at 4 wt.% silane (sample 4), then dived to 1.55% abruptly at 6 wt.% silane (sample 5), and finally decreased gently to 1.25% at 8 wt.% silane (sample 6). On the one hand, after NaSi treatment, the surface compatibility was enhanced, resulting in an increasing of crystalline. On the other hand, some oligomer or biopolymer produced by the auto polymerization of siloxane served as a lubricant of PLA, leading to the improvement of PLA molecular mobility and the decline of crystalline.

The thermal stability of the PLA/WSP biocomposites was investigated while using TG analysis and it is shown in Figure 9. It can be seen that all types of composites embodied a similar trend (single step degradation) during the whole temperature range. PLA possessed the highest onset temperature of degradation (T1) of 291.04 °C and the temperature at maximum degradation rate (T2) of 353.5 °C.

The incorporation of the WSP into the PLA matrix dramatically reduced the T1 and T2 to 239.87 °C and 304.9 °C. Biocomposites with NaOH/KH550 treated WSP behaved better with a higher thermal stability with T1 of 264.87–276.37 °C and T2 of 306.8–327.3 °C as compared to the PLA biocomposites with the untreated WSP. Among those, T1 and T2 of biocomposites with 8 wt.% silane treated WSP reached their highest value.

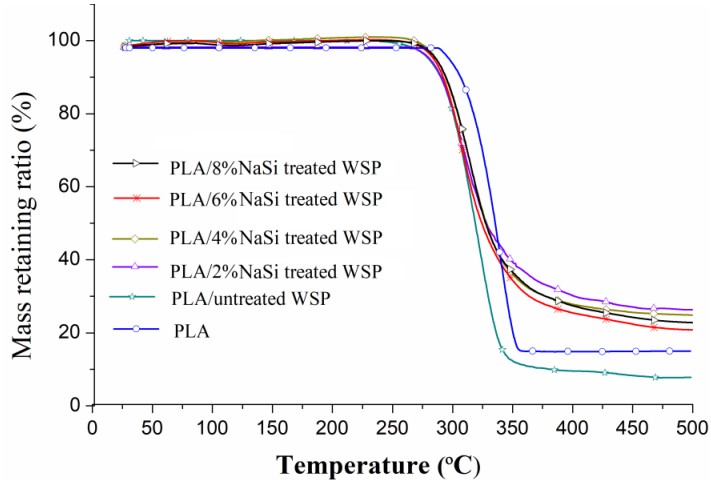

**Figure 9.** Thermo Gravimetric (TG) curves of PLA/WSP biocomposites.

The WSP consisted of 13.21% of cellulose, 55.24% of hemicellulose, and 26.79% of lignin and 5% of other impurities [46]. During the thermal degradation, the temperature of mass loss of hemicelluloses, cellulose, and lignin was at a range of 220–315, 315–400, and 160–900 °C, respectively [46]. Therefore, the thermal stability of hemicelluloses and lignin containing WSP/PLA composites became lower than that of neat PLA. The NaOH removed a certain amount of lower thermally-stable substances, such as hemicelluloses, oil, and waxes from the surface of WSP (Figure 6), leading to the increase of the degradation temperature. However, the lignin still consisted more than 50% of the WSP, so that the degradation temperature of the PLA composites with the treated WSP was lower than that of PLA. Chemical reaction between the silane and the WSP, and the physical deposition of the silane on the WSP limited the molecular mobility and enhanced the molecular stability of PLA, leading to a more stable thermal property. However, when the content of KH550 continuously increased, the number of free molecular chains of siloxane was increasing, as a result of the drop in thermal stability.

After the maximum degradation, the pure PLA and the PLA/WSP biocomposites had no almost residue. Biocomposites with the treated WSP possessed 22.3–28.1% of the residual mass. The 2 wt.% silane treated WSP composites had the highest residual mass. The main components of the WSP were cellulose and lignin after the alkali/silane treatment. The treatment of WSP with the silane slowed down the degradation of lignin, which continued to degrade under higher temperatures.

The thermal properties indicated that silane treatment can enhance the thermal properties of PLA/WSP biocomposites. The similar effect of silane on natural fibers happened on polyester/coconut fiber composites (0.5 wt.% silane) [24], PLA/coir fiber composites (0.5–1 wt.% silane) [47], and PLA/corn composites (2–3 wt.% silane) [48].

*3.4. Mechanical Properties Analysis of Specimens Fabricated with FDM*

3.4.1. The Effect of Silane Treatment on the Mechanical Properties of the FDM PLA/WSP Biocomposite Samples

Table 1 shows the average mechanical properties of FDM PLA, untreated and treated PLA/WSP biocomposite samples. All of the compression samples had a high plasticity and could not be crushed, even under a pressure of 50 KN. Accordingly, the compressive strength was recorded with a compression

ratio of 67% for safety. From Table 1, it can be seen that the pure PLA almost possessed the highest strength and modulus at all items, except the tensile modulus. There was a decline in the mechanical properties for PLA/untreated WSP as compared to the pure PLA (sample 2). Its compressive strength and modulus dropped drastically by 59.8%. Its flexural strength considerably decreases by 36.4%. Whereas its tensile strength and flexural modulus performed more moderate with a drop from 57.3 MPa and 35.8 MPa to 48.3 MPa and the 33.5 MPa, respectively. However, the tensile modulus had a dramatic climb from 1908.7 to 3050.4 MPa, indicating that the capacity of anti-tensile deformation was enhanced.

**Table 1.** The mechanical properties of Fused Deposition Modeling (FDM) PLA, untreated, and treated Poly (lactic acid) /Walnut shell powder (PLA/WSP) biocomposite samples.

| Samples | Tensile Strength (MPa) | Tensile Modulus (MPa) | Flexural Strength (MPa) | Flexural Modulus (MPa) | Compressive Strength (MPa) | Compressive Modulus (MPa) |
|---|---|---|---|---|---|---|
| pure PLA | 57.3 ± 1.0 | 1908.7 ± 9.4 | 69.0 ± 1.1 | 35.8 ± 1.2 | 526.9 ± 3.6 | 65.9 ± 0.9 |
| PLA/untreated WSP | 48.3 ± 0.5 | 3050.4 ± 25.1 | 44.1 ± 0.6 | 33.5 ± 2.5 | 212.1 ± 1.5 | 26.5 ± 0.5 |
| PLA/2%NaSi treated WSP | 49.6 ± 1.2 | 2284.5 ± 5.25 | 47.1 ± 0.6 | 32.3 ± 0.8 | 219.9 ± 1.3 | 27.5 ± 0.2 |
| PLA/4%NaSi treated WSP | 50.8 ± 1.2 | 2442.7 ± 16 | 49.9 ± 0.8 | 32.1 ± 0.9 | 252.5 ± 2.1 | 31.6 ± 0.3 |
| PLA/6%NaSi treated WSP | 50.3 ± 0.1 | 2381.5 ± 15.7 | 51.2 ± 0.9 | 31.17 ± 1.07 | 266.82 ± 1.22 | 33.35 ± 0.15 |
| PLA/8%NaSi treated WSP | 52.5 ± 0.7 | 2570.5 ± 22.8 | 51.7 ± 2.8 | 28.8 ± 3.1 | 276.6 ± 2.9 | 34.6 ± 0.7 |

There are three factors that might affect the mechanical properties. Firstly, the WSP has a higher amount of lignin, lower amount of cellulose and hemicellulose, and a small amount of ash, oil, and other impurities. The higher amount of lignin aggravated the brittleness of composites [49], and the ash had a negative effect on the bonding [43], leading to a lower strength. Secondly, the rigid WSP particles boosted the stiffness of the PLA/WSP composite, and a small number of Nano-scale WSP particles acted as a nucleating agent of PLA [24], both resulting in the improvement of the mechanical strength and modulus. In this research, 5 wt.% content of the untreated WSP particles might be too much for the polymer matrix of PLA, because the effect of hemicelluloses and other impurities on mechanical properties was stronger than its function as a nucleating agent. Thirdly, FDM is a kind of layer-by-layer manufacturing, so the impact of the integration degree between layers on the mechanical properties cannot be ignored [35]. The manufacturing direction was identical with the compressive direction, and perpendicular to the tensile axis and the flexural center. Therefore, the bonding degree between layers played a more significant role in compressive properties than in flexural properties. It resulted in a dramatic decline of compressive property and a moderate drop of flexural property.

For improving the compatibility of PLA and WSP, the WSP was modified by the NaOH and the silane (2, 4, 6, and 8 wt.% concentration). As shown in Table 1, the tensile and flexural strength of PLA/NaSi treated WSP showed a slightly rising trend with an increment of 4.2 MPa and 6.9 MPa for each, when the concentration of KH550 was increased, as compared to PLA/untreated WSP. The PLA/8%NaSi treated WSP (sample 6) gets maximum values of tensile strength (52.5 MPa) and flexural strength (51.7 MPa). However the modulus of PLA/NaSi treated WSP experienced a thoroughly negative performance and decreased slowly to 2570.5 MPa of tensile modulus (sample 6) and 28.8 MPa of flexural modulus (sample 5), respectively. The compressive strength PLA/NaSi treated WSP grew slightly to 219.9 MPa at 2 wt.% silane following a rapid rise to 252.5 MPa at 4 wt.%, and then continuously increased to the highest value of 276.6 MPa at 8 wt.% silane. The compressive modulus showed a steadily upward tendency when the silane concentration was increased, and gradually leveled off to a peak of 34.6 MPa at 8 wt.% silane.

As previously analyzed, the alkali removed the hemicelluloses and impurities from the WSP, so that the molecular chains of the WSP rearranged and repacked themselves. The deposition of auto polymerized siloxane on the micro pores and the surface of WSP, which enhanced the interfacial compatibility between the PLA and the WSP. Therefore, during this FDM process, plastic melts between two scan lines or two layers enabled an improved fusion and adhesion, so that more energy was needed to resist the mechanical force. In general, the overall mechanical performance of PLA/untreated WSP was improved through NaOH/KH550 treatment and was optimized at 8 wt.% KH550.

3.4.2. The Effect of WSP Weight Ratio on the Mechanical Properties of FDM PLA/WSP Biocomposite Samples

The mechanical properties of FDM PLA biocomposites with WSP treated with 8 wt.% concentration of silane are better than that of treated with other concentration of silane. Therefore, the PLA/8%NaSi treated WSP biocomposite filament was chosen to develop farther experiments. Table 2 gives the tensile properties of the FDM samples of PLA biocomposites with 5–15 wt.% WSP treated with 8 wt.% concentration of silane. When increasing the ratio of WSP, the tensile strength of biocomposite samples was improved from 52.5 MPa to 56.2 MPa at 10 wt.% WSP, followed by a moderate drop to 55.3 MPa at 15 wt.% WSP. The silane coupled reaction with both PLA and WSP resulted in a high interfacial bonding [48]. The good interfacial compatibility was beneficial to transfer stress in the tensile test. The result is similar to that in references for PLA/corn fibers composites [48] and Polypropylene/Argan nut shells composites [50]. The tensile modulus dropped down by 19.1% to 2080.5 MPa at 10 wt.% WSP and increases a little to 2177.9 MPa at 15 wt.% WSP, which indicates that the anti-deformation capability was higher for a higher ratio of WSP. The improvement showed that a higher ratio of WSP was beneficial to improve the mechanical properties of the FDM PLA/WSP biocomposite samples. The biocomposite filaments have the potential to be used as cheaper FDM materials than the PLA filaments. The performance of the tensile modulus was similar to that in the research of the PLA/coir fiber composites [45].

**Table 2.** Mechanical properties of Fused Deposition Modeling (FDM) Poly (lactic acid) /Walnut shell powder (PLA/WSP) biocomposite samples.

| Samples | Tensile Strength (MPa) | Tensile Modulus (MPa) |
|---|---|---|
| PLA/5 wt.%WSP | 52.5 ± 0.7 | 2570.5 ± 22.8 |
| PLA/10 wt.%WSP | 56.2 ± 0.6 | 2080.5 ± 10.8 |
| PLA/15 wt.%WSP | 55.3 ± 0.3 | 2177.9 ± 19.4 |

*3.5. Morphological Analysis of Untreated and Treated Samples Fabricated with FDM*

Figure 10 shows the morphology of pure PLA, PLA/untreated WSP, and PLA/NaSi treated WSP, taken from the tensile fractured surface of FDM samples. The pure PLA specimen shows a stair-step fractured surface with some cracks, indicating a full brittle fracture with some plastic features (Figure 10a). The fracture surface of PLA/untreated WSP (Figure 10b) is smoother than that of pure PLA, implying a full brittle fracture. The WSP particles in Figure 10b were partially aggregated. The micro voids in Figure 10b resulted from the pull-out of WSP particles during the tensile process. The voids and aggregated particles both revealed that there was small energy consumption during the pulling-out of the particle. The leaks between the two layers were obvious and there was no apparent deformation that happened during the tensile test. This phenomenon showed that the introduction of the WSP particles reduced the flow of polymer and lowered the ductility.

Figure 10c–f show the effect of NaOH/KH550 on the morphology of the PLA/WSP biocomposite samples. It can be observed that the number and the size of the voids and defects decreased remarkably with the rise of the KH550 content. In Figure 10c, the surface of voids is rougher than those in Figure 10b. The matrix around the voids was extended when the WSP particles were pulled out. There was a small gap between particles and matrix. This situation implied that there was higher energy consumption during the particles pull-out and a more interfacial compatibility between fillers and matrix. A slight deformation of leaks between layers showed enhanced plastic fluidity. With the increasing of KH550 content, the voids became smaller in Figure 10d and gradually disappeared in Figure 10e,f; the WSP particles dispersed more evenly (Figure 10d) than those in Figure 10c; the extension degree of polymers around voids and WSP particles became stronger (Figure 10d,e,g), until it dispersed on almost the whole surface (Figure 10f,h). When increasing the ratio of treated WSP, the leak in surface was dramatically decreased (Figure 10g) until it completely disappeared (Figure 10h);

the voids due to the particle pull-out became very shallow; and, the WSP particles were integrate as a whole with the PLA matrix in Figure 10h, showing a good adhesion between WSP and PLA matrix. The results showed the effect of NaOH/KH550 on the improvement of the particle dispersion and the surface compatibility, meaning an enhancement of heat-resistance and mechanical properties.

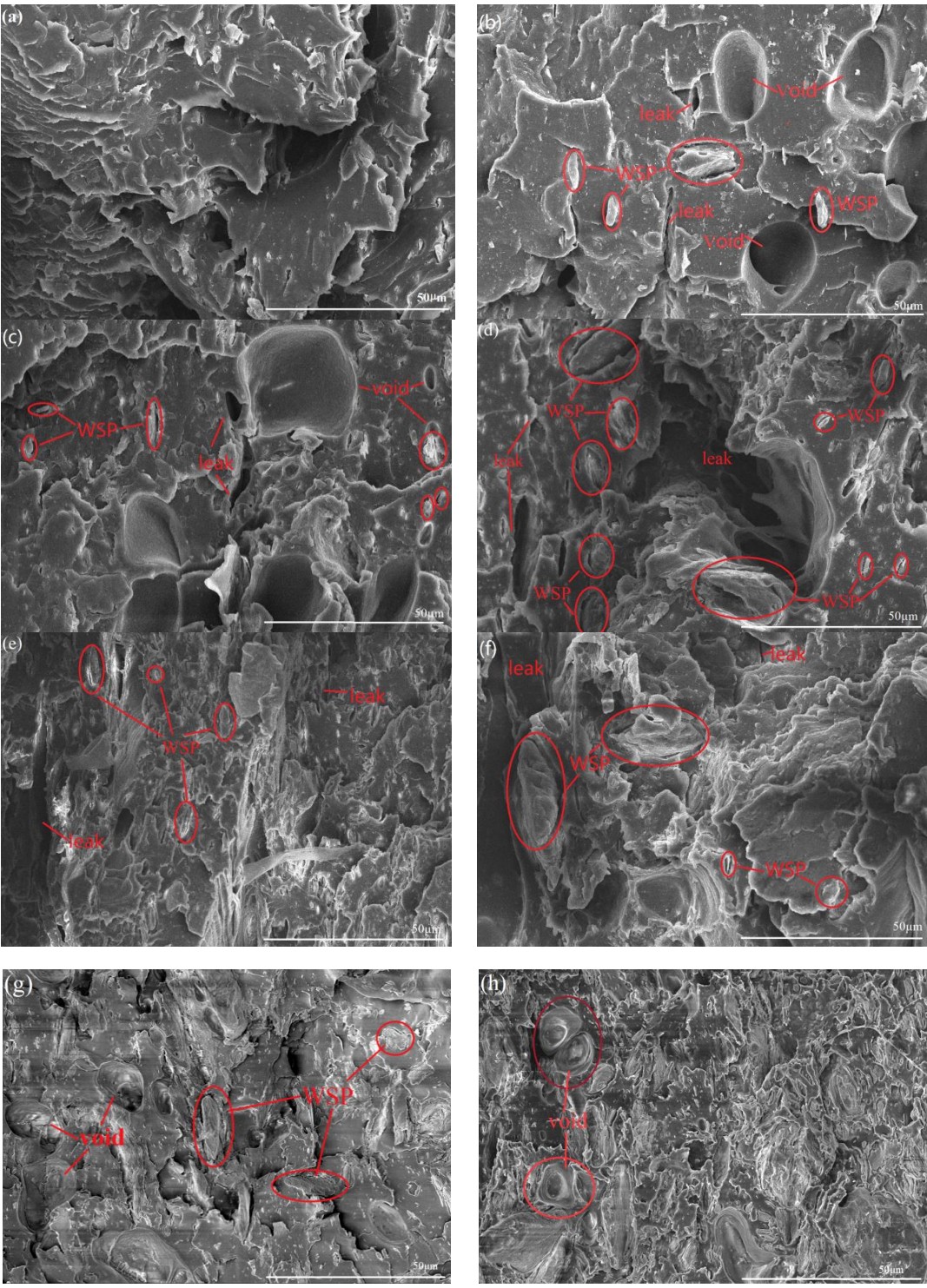

**Figure 10.** The scanning electron microscopy (SEM) morphology of FDM samples: (**a**) pure PLA, (**b**) PLA/untreated WSP, (**c**) PLA/2%NaSi treated WSP, (**d**) PLA/4%NaSi treated WSP, (**e**) PLA/6%NaSi treated WSP, (**f**) PLA/8%NaSi treated WSP, (**g**) PLA/10 wt.% treated WSP, and (**h**) PLA/15 wt.% treated WSP.

### 3.6. Fabrication of Porous Scaffolds Using FDM

Given that the FDM PLA/15 wt.% WSP biocomposite possessed a good mechanical property in this study, it was chosen to further fabricate porous scaffolds with various porosities (40–75%) and interconnected pores in sizes of 50–200 μm. It can be seen from Figure 11, with an increasing porosity, the yield strength and modulus experienced a drop, getting to a highest value of 19.96 MPa and 104.1 MPa at 43.43% of porosity; and, getting to a minimum value of 1.67 MPa and 10.93 MPa at 74.41% of porosity, respectively. The results showed that the scaffolds with higher porosity would have reduced strength and modulus. The strain went a contrary route, as compared to the strength, climbing sharply to its peak (25.96%) at 64.17% of porosity, followed by a sharp drop to its minimum (15.28%) at 74.41% of porosity. It deserves to be observed that the scaffold with porosity of 64.17% possessed the highest strain, a decent strength and modulus, showing an excellent ability to sustain the compression load.

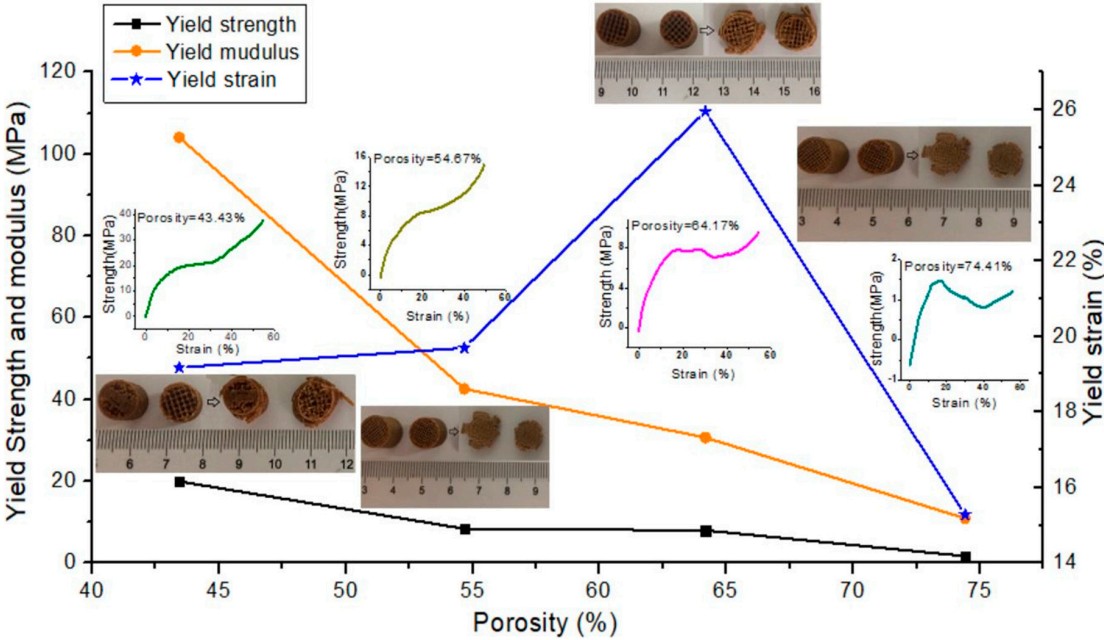

**Figure 11.** The relationship between porosity and compression properties.

The compression process included two stages, the yield plateau and densification. The width of yield plateau played a key role in deciding the ability of compression energy absorption. The scaffold with porosity of 64.17% possessed the biggest yield plateau width, so that it could not be crushed, even at the deformation of 55%. However, the plateau width decreased when increasing the porosity to 74.41%.

From the picture of the scaffold before compression, it can be observed that the pores were distributed between 50–200 μm. The result showed that not only the porosity, but also the pore size, could be controlled through FDM. The results in the literature revealed that the compression modulus of a trabecular bone was 10 to 2000 MPa [51,52]. If the size of the pores is in the range of 10–75 μm, the fibrous tissue was able to penetrate into the pores of the scaffold. If the range is 75–100 μm, it led to the ingrowth of un-mineralized osteoid tissue. If the size was larger than 200 μm, it resulted in substantial bone ingrowth [51,52]. The higher porosity and larger interconnected pores benefit the bone ingrowth and osseointegration of implants after surgery [35]. Therefore, in this work, the scaffold with porosity of 40–75%, the pore size in the range of 50–200 μm, and the compression modulus of 10–104 MPa can satisfy the requirements of the trabecular bone implants, and it has the potential to be used in tissue engineering. The incorporation of 15 wt.% WSP can cut down the cost of PLA by about 15% and enhance the degradation speed of PLA.

## 4. Conclusions

The present study showed that FDM can be successfully utilized to process relatively cheap PLA/WSP biocomposites filaments and achieve acceptable thermal and mechanical properties. The experimental results showed: the thermal and mechanical properties of PLA/untreated WSP decreased due to the poor compatibility between WSP and PLA, which was dramatically improved by NaOH/KH550 treatment; the crystalline, thermal gravity, and thermal stability of untreated biocomposite was improved from 1.46%, 60. 3 °C, and 239.87 °C to 2.84%, 61.3 °C, and 276.37 °C; the tensile, flexural, and compressive strength mounted to 52.5 MPa, 51.7 MPa, and 276.6 MPa, respectively. The tensile strength was further improved to 56.2 MPa by adding more WSP into PLA matrix. SEM morphology revealed that the number and size of micro voids, and leaks between layers were reduced with an increase of the KH550 concentration and ratio of WSP. Finally, the porous scaffolds with controllable porosity (40–75%) and pore size (50–200 μm) were manufactured. 10.93–104.07 MPa of the compression modulus showed that the scaffold has the potential to be used as trabecular bone implants applied in tissue engineering. The utilization of FDM technology employing walnut shells in the process not only cut down the cost of PLA filament by about 15%, but also led to several benefits, such as making good use of wastes, enhancing the degradation of PLA, and customizing complex components with interconnected pores and suitable porosity.

**Author Contributions:** Conceptualization, X.S. and W.H.; methodology, S.Y.; software, G.H.; validation, X.S. and W.H.; formal analysis, X.S.; investigation, X.S. and T.Y.; data curation, X.S.; writing—original draft preparation, X.S.; writing—review and editing, W.H. and S.Y.; visualization, G.H.; supervision, W.H.; project administration, X.S. and W.H.; funding acquisition, X.S.

**Funding:** This research was funded by "(1) the National Nature Science Foundation of China, grant number 51705094", "the Nature Science Foundations of Guangxi province, grant numbers 2017JJB150072 and 2018JJA160218".

**Acknowledgments:** The authors gratefully acknowledge financial support from: the National Nature Science Foundation of China (51705094), the Nature Science Foundations of Guangxi province (2017JJB150072 and 2018JJA160218).

**Conflicts of Interest:** The authors declare no conflict of interest.

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
