# Peer review of "Fused Deposition Modeling of Poly (Lactic Acid)/Walnut Shell Biocomposite Filaments—Surface Treatment and Properties"

_applsci, doi:10.3390/app9224892_

Round 1
Reviewer 1 Report
In short: the article shows an interesting approach to creation of a composite filament suitable for use in 3D printing. While there is a new filling material (walnut shell) and a decent part of the research is devoted to creation of strong adhesion between the polymer and the filler, the goal is not well defined (i.e. if the goal was to reduce the content of relatively cheap PLA in the filament while pertaining acceptable mechanical properties of printed parts, or if it was designed to exceed mechanical properties of current composites).
Recommendation: major revision to comply with proper research design
applied science review
title
The title is confusing, it suggests that the biocomposites are subject to surface/thermal treatment, however they are not. The treatment is applied to one component that will constitute the composite (the walnut shell) prior to combining it with PLA to produce the composite.
Such misleading facts could be found everywhere in the paper. For example:
line 70 “firstly attempted to fabricate PLA/WSP biocomposites with FDM”:
The composite is not fabricated with FDM, it was made with a single screw extruder which produced the composite filament. The filament, in turn, is suitable to be used as material for Fused Filament Fabrication.
20-21 I would recommend to avoid “most optimum” word construction. Optimal is “the most” by definition.
110 - how the moisture content was measured?
115 - what is the size ratio? Average particle A size to the average particle B size? 300-700 to 20-45 is not 10:1. Why 10:1 is beneficial?
125 what ratio?
129 Maybe: Preparation of filament for FDM?
135 What were the criteria for optimization?
148 Same question. Authors are overusing all forms of term “optima”. There is a huge difference between “workable” and “optimal”. Optimisation means the process of finding the best solution using a predefined loss function until any modification of the parameters leads to worse results, using one of the well-known optimization strategies and algorithms.
150-152 What is the shell thickness and infill pattern of 3D printed samples? What slicer was used?
153 I am not familiar with GBT 9341 2 008 standard, but ASTM 638 prescripts to cut the samples out of monolithic material or to inject them into molds. Fabrication by 3D printing is not mentioned in the standard.
364-380 The mechanical properties description could be supplemented with the experiment on the influence of adding certain percentage of WSP into PLA on the tensile strength, compared in any fixed printing mode
The article requires rigorous editing.
There are no estimations of the economical effect of creating the composite, even rough cost comparison with pure PLA filament.
Any conclusions that the material proposed is applicable for «medical bones» is not supplemented with any kind of studies on biocompatibility of the new material. That research lies in the biomedical area indeed and must comply with standards by FDA or similar organization.
Author Response
Response letter for reviewer 1
Dear Reviewer:
Thanks very much for your time and efforts, all your professional points have been revised in paper and responded as follows:
The goal of the paper is not well defined.Response: The goal of the paper was to 3D printed relatively cheap PLA biocomposites filaments with acceptable mechanical properties by incorporation of WSP, shown in line 72-73 and line 464 in blue.
The title is confusing.Response: The title has been changed as “The influence of the surface treatment of Walnut shell on the properties of 3D printed Poly (lactic acid)/Walnut shell biocomposites”.
Some misleading facts, for example, in line 70?Response: These kind of misleading facts have been revised by using word “filament” or change the sentence structure. Please check in line 2-4, line 16, line 72-73 and line 465 in blue.
I recommend to avoid “the most optimum” word construction.Response: This kind of expression has been revised to others, for example, determined, shown in line 24, line 78, line 156 and line 373 in blue.
How to moisture content was measured?Response: The moisture content was measured by using a scale with an error of 0.001. Weigh the drying materials every 3h until there is no change in the weight shown in line 108 in blue.
What is the size ratio? Average particle A size to the average particle B size? 300-700 to 20-45 is not 10:1. Why 10:1 is beneficial?Response: The size ratio is “the average particle A size to the average particle B size”. There was a literature proved that “The particle size ratio between the matrix material and the filler should be 10:1 or over, which is beneficial to the homogeneity of the mixture”. In this paper, the particle ratio is more than 10:1, which is satisfied the requirement. The relative content is in line 123-124 in blue.
What ratio?Response: The ratio is weight ratio. Please see in line 133 in blue.
Maybe: Preparation of filament for FDM?Response: The headline has been revised according to your recommendation, shown in line 136.
What were the criteria for optimization?Response: According to the thermal properties of materials and the results of trial and error, the parameters can be determined. Please check in line 142 in blue.
The overusing of optima?Response: This kind of expression has been revised to others, for example, determined, shown in line 24, line 78, line 156 and line 373 in blue.
What is the shell thickness and infill pattern of 3D printed samples? What slicer was used?Response: The shell thickness was 0.4mm, which is equal to the diameter of nozzle (line 157). The infill pattern was linear filling mode. The slicer was the Allct software, which slice a part and create a G.code for 3D printing (line 158).
About “the fabrication by 3D printing is not mentioned in the standard ASTM 638?Response: We didn’t find the standard for 3D printed tensile samples, so we have to refer to some relevant standard.
The mechanical properties description could be supplement with the experiment on the influence of adding certain percentage of WSP into PLA on the tensile strength, compared in any fixed printing mode?Response: According to your recommendation, the experiment and analysis of 3D printed PLA/10-15 wt.% WSP bicocomposites have been carried out, shown in 382-398, line 420-424 and line 430 in blue. After the properties tests of PLA biocomposites with 10wt.% and 15wt.% treated WSP, the latter was chosen to fabricate porous scaffold. The original material for scaffold was PLA biocomposites with 5wt.% treated WSP, shown in line 435-463 in blue.
The article requires rigorous editing.Response: Done as suggested.
Any conclusions that the material proposed is application for《medical bones》is not supplemented with any kind of studies on biocompatibility of the new material. That research lies in the biomedical area indeed and must comply with standards by FDA or similar organization.Response: The two materials used in this paper all can be used in medical. PLA derived from renewable resources, is a biodegradable, recyclable, and compostable thermoplastic with good biocompatibility, has been extensively used in medical. The literature “T. Patrı´cio et, al. Fabrication and characterisation of PCL and PCL/PLA scaffolds for tissue engineering. Rapid Prototyping Journal, 2014” used PLA 2002D as its experiment material. Walnut is a kind of biodegradable and recyclable natural material. A health claim authorized by FDA indicated that walnuts can reduce the risk of heart disease (line 42).
Extensive editing of English language and style required.Response: Done as suggested.

Reviewer 2 Report
The authors presents a study on the effects of surface treatment of PLA/WSP.
The topic is well introduced, and the procedures and conclusions are relevant and well explained.
I consider this work is acceptable for publication with the consideration of some minor language editing (i.e. in line 71: the purpose was to investigate).
Author Response
Response letter for reviewer 2
Dear Reviewer:
Thanks very much for your time and efforts, all your professional points have been revised in paper and responded as follows:
There are some minor language editing in the paper, please improve it.Response: Done as suggested.

Reviewer 3 Report
In the manuscript, the author demonstrates PLA-walnat shell particle composites for 3D printing. The authors estimated their mechanical properties and thermal stability. However, the reviewer decided to give major revision for following reasons with comments as follows,
1) As a study on composite materials for 3d printing, authors need to specify the concentration of WSPs in the composite materials. If the concentrations are not fixed, it is not conclusive that surface treatment of WSP can result in improved mechanical property and interface compatibility. The reviewer suggests to measure their mechanical property by changing wt.%.
2) In row 46, the author mentions about the hydrophobicity of polymer and incompatibility. In Figure 1, the author shows a WSP after cutting. Does this show particles after impurity removal? If not, author may need to present clean particles.
3) In row 224, the author mentions "However, the Si O C bonds didn’t show up, implying that there was no obvious reaction between OH groups of WSP and hydrolyzed silanes." The author may need to further clarify if the silane coupling agent is physically adsorbed at the interface or not. If it is physically adsorbed, can further washing remove all of the silane coupling agents? Also, can the author still mention the importance of chemical treatment as is shown in the introduction? These are contradicted each other. If it is difficult to see it with IR spectra, it is recommended to use XPS.
4) In the interpretation part of DSC data, it is advisable if the author can explain better about the relation among molecular mobility, nucleation and interfacial compatibility between fillers and polymer matrix with proper reference. Also, more interpretations based on chemical structures of silane, polymer, and WSP are needed.
5) For application to 3D printing, one of the key factor is the rheology of the plastic ink. The authors need to perform rheology measurement to see if there are any clue of jamming at the nozzle which happens often with particle reinforced 3D printing materials.
6) In the TGA data, it is advisabl to show TGA of pure WSP and treated WSPs. Without these control experiments, it is difficult to imagine the degradation temperature of the composites. The raw WSPs contain lots of impurities as the author mention, and this can affect on the TGA data significantly. Also, in the TGA data, the reviewer did not see the correlation between the amount of silane and the wt. ratio of the composite after the experiment. the authors need to describe what is the main reason for this.
7) In row 278, the author mentions "Lower degree of crystalline means lower thermal stability." The author may need to further address how decomposition temperature is related to the crystallinity. This needs more explanation.
8) In row 287, the author mentions "The thermal properties indicated that a suitable concentration of silane (6-8wt.%) could enhance the thermal properties of PLA/ASP biocomposites." This sentence needs more explanation.
When authors address all of these issues, the manuscript will be reconsidered.
Author Response
Response letter for reviewer 3
Dear Reviewer:
Thanks very much for your time and efforts, all your professional points have been revised in paper and responded as follows:
As a study on composite materials for 3d printing, authors need to specify the concentration of WSPs in the composite materials. If the concentrations are not fixed, it is not conclusive that surface treatment of WSP can result in improved mechanical property and interface compatibility. The reviewer suggests to measure their mechanical property by changing wt.%.Response: (1) The concentration of WSP was fixed as 5wt.% in order to investigate the effect of different concentration of silane on properties of PLA/WSP biocomposites, shown in line 129 in blue. (2) With the optimized concentration of silane, the effect of content of WSP in the PLA biocomposites on the tensile properties and surface morphology was carried out, shown in line 383-399, line 420-424 and line 430 in blue. (3) After the properties tests of PLA biocomposites with 10wt.% and 15wt.% treated WSP, the latter was chosen to fabricate porous scaffold. The original material for scaffold was PLA biocomposites with 5wt.% treated WSP, shown in line 435-463 in blue.
In row 46, the author mentions about the hydrophobicity of polymer and incompatibility. In Figure 1, the author shows a WSP after cutting. Does this show particle after impurity removal? If not, author may need to present clean particles.Response: Fig. 1 in the original paper was the picture of WSP without treatment by NaOH/silane. Therefore, according to your recommendation, a new picture of WSP which had been treated by NaOH/silane, please check in line 111-120 in blue.
In row 224, the author mentions "However, the Si O C bonds didn’t show up, implying that there was no obvious reaction between OH groups of WSP and hydrolyzed silanes." The author may need to further clarify if the silane coupling agent is physically adsorbed at the interface or not. If it is physically adsorbed, can further washing remove all of the silane coupling agents? Also, can the author still mention the importance of chemical treatment as is shown in the introduction? These are contradicted each other. If it is difficult to see it with IR spectra, it is recommended to use XPS.Response: Although Si-O-C groups didn’t show up obviously, there was still a chemical reaction (condensation) between WS and silane, shown in 236-238. There was also a physical intertangling due to the auto polymerization of silanol groups deposited on the surface of WSP and went into the micro pores of WSP, shown in line 238-241. Compared to the PLA biocomposite with untreated WSP, the thermal stability and the mechanical properties of the PLA biocomposite with NaOH/silane treated WSP possessed higher mechanical properties. The results showed that the silane coupling agent was not been removed and still on the WSP particles when blended with PLA. Therefore, the authors can mention the importance of the chemical treatment in the introduction.
In the interpretation part of DSC data, it is advisable if the author can explain better about the relation among molecular mobility, nucleation and interfacial compatibility between fillers and polymer matrix with proper reference. Also, more interpretations based on chemical structures of silane, polymer, and WSP are needed.Response: Done as suggested, shown in line 278-280, line 284 and line 292-295 in blue.
For application to 3D printing, one of the key factors is the rheology of the plastic ink. The authors need to perform rheology measurement to see if there is any clue of jamming at the nozzle which happens often with particle reinforced 3D printing materials.Response: The rheology test had been supplemented, shown in line 197-203 and line 246-260 in blue.
In the TGA data, it is advisable to show TGA of pure WSP and treated WSPs. Without these control experiments, it is difficult to imagine the degradation temperature of the composites. The raw WSPs contain lots of impurities as the author mention, and this can affect on the TGA data significantly. Also, in the TGA data, the reviewer did not see the correlation between the amount of silane and the wt. ratio of the composite after the experiment. The authors need to describe what the main reason is for this.Response: Done as suggested, shown in line 316-318 and line 328-332 in blue.
In row 278, the author mentions "Lower degree of crystalline means lower thermal stability." The author may need to further address how decomposition temperature is related to the crystallinity. This needs more explanation.Response: This statement has been removed and was replaced by others, shown in line 316-323.
In row 287, the author mentions "The thermal properties indicated that a suitable concentration of silane (6-8wt.%) could enhance the thermal properties of PLA/ASP biocomposites." This sentence needs more explanation.Response: This statement has been removed was replaced by others, shown in line 333-334.
Extensive editing of English language and style required.Response: Done as suggested.
Reviewer 4 Report
The subject of the research described in the article entitlend Effect of surface treatment on the thermal and mechanical properties of poly(lactic acid)/Walnut shell bio-composites fabricated by 3D printing by Xiaohui Song, Wei He, Shoufeng Yang, Guoren Huang, Tonghan Yang is the production of bio-composite from powder poly(lactic acid)/walnut (PLA/WSP) using 3D printing.
Many tests were carried out to obtain the most effective filler in the form of ground walnut shells as part of the research. The thermal and mechanical properties of the obtained bio-composite have been optimized by adjusting the silane concentration. Valuable materials have been obtained that can be used in tissue engineering.
I have one remark to the Authors: Some sentences (and there are quite a lot of them in the whole text) are written in a string, without spaces. Please correct this in the text..
Apart from this one remark, I think that the article represents a high scientific level (both in terms of content and editing). I think, that this article could be publish in the present form.
Author Response
Response letter for reviewer 4
Dear Reviewer:
Thanks very much for your time and efforts, all your professional points have been revised in paper and responded as follows:
Some sentences (and there are quite a lot of them in the whole text) are written in a string, without spaces. Please correct this in the text.Response: Done as suggested.

Round 2
Reviewer 1 Report
I am mostly satisfied with the improvements introduced to the article.
I now can recommend the editorial board to accept the article for publication, however I have to draw authors attention to some issues.
Title:
The biocomposite is material and this material has not been 3D printed. Samples (objects) were 3D printed out of new material and then tested to get a clue on influence of surface treatment of one of the biocomposite component to the properties of 3D printed parts. I suggest to elaborate the formulation so it is clear that there is a part printed of such composite in question.
Lines 15-16
Biocomposite filaments are suitable to be used in FFF 3D printing, as proven by the research, but biocomposite filaments were not fabricated using 3D printing technique.
Same misleading phrases can be found on lines 72 and 76.
Line 67. I would not agree with statement that FDM == 3D printing. I would recommend to use standard terminology: https://web.mit.edu/2.810/www/files/readings/AdditiveManufacturingTerminology.pdf
Lines 97-98. Connection between text and reference [11] is missing.
Mainly it is a language issue, but I am afraid that many more issues like these are buried in the text. I strongly recommend to apply rigorous post-editing by a native English speaker with some understanding of the research area.
Thank you for your attention and good luck with final approach on this particular article and future researches!
Author Response
Response letter for reviewer 1
Dear Reviewer:
I really appreciate your professional opinion about my paper. After being twice revised, this paper has improved dramatically. I have built a good understanding about AM, 3D printing and FDM.I believe in the future, I will be able to use the correct description to depict different kinds of materials, such as filaments, samples and biocomposite powders.
You are very professional in this field, thank you so much again for being our reviewer, and I look forward to meeting you in the future.
Thanks very much for your time and efforts, all your professional points have been revised in paper in the following way:
About the title.Response: The title has been changed to “Fused deposition modeling of Poly (lactic acid)/Walnut shell biocomposite filaments—surface treatment and properties”.
Line 15-16: Biocomposite filaments are suitable to be used in FFF 3D printing, as proven by the research, but biocomposite filaments were not fabricated using 3D printing technique.Response: Definitely, you are right about the understanding of the FFF and 3D printing. The incorrect description has been revised, shown in line 15-16 in green.
Same misleading phrases can be found on lines 72 and 76.Response: The misleading phrases have been revised, shown in line 72-75 in green.
Line 67. I would not agree with statement that FDM == 3D printing. I would recommend to use standard terminology: https://web.mit.edu/2.810/www/files/readings/AdditiveManufacturingTerminology.pdfResponse: Thank you very much for the sharing of the AM terminology. All the misleading phrases have been revised, shown in line 27-28, 67-68, 156, 165, 340-342, 361, 382, 387-388, 392, 401-402, 406 in green.
Line 97-98.Connection between text and reference [11] is missing.Response: I am so sorry for this mistake, which has been removed.
Mainly it is a language issue, but I am afraid that many more issues like these are buried in the text. I strongly recommend to apply rigorous post-editing by a native English speaker with some understanding of the research area.Response: The language and the grammar have been improved by a native English speaker, who works in my university and has published several SCI papers in the field of mathematics. During the urgent time (last time, 13 days for a major revise and this time, only 1 day for a minor revise), I could not find a person who comes from an English speaking country, and meanwhile is professional in this field. So, my colleague and I looked into the paper again to try to improve it.
Hopefully, the improvement of the paper can satisfy your professional requirements.
Thank you!
